# Investigation on AC and DC Breakdown Mechanism of Surface-Ozone-Treated LDPE Films under Varied Thicknesses

**DOI:** 10.3390/polym15234490

**Published:** 2023-11-22

**Authors:** Yongjie Nie, Jie Liu, Junxin Ke, Xianping Zhao, Shengtao Li, Yuanwei Zhu

**Affiliations:** 1Electric Power Research Institute, Yunnan Power Gird Co., Ltd., Kunming 650217, China; 2State Key Laboratory of Electrical Insulation and Power Equipment, School of Electrical Engineering, Xi’an Jiaotong University, Xi’an 710049, China

**Keywords:** space charge, electrical breakdown, surface oxidation, thickness dependent

## Abstract

Electrical breakdown is an important physical phenomenon in power equipment and electronic devices. Recently, the mechanism of AC and DC breakdown has been preliminarily revealed as electrode–dielectric interface breakdown and bulk breakdown, respectively, based on space charge dynamics through numerical calculations. However, the AC breakdown mechanism still lacks enough direct experimental support, which restricts further understanding and the design and development of electrical structures. Here, in this study, LDPE films with various thicknesses ranging from 33 μm to 230 μm were surface modified with ozone for different durations to experimentally investigate DC and AC breakdown mechanism. The results indicate that carbonyl groups (C=O) were introduced onto the film surface, forming shallow surface traps and leading to a decreased average trap depth and an increased trap density. Such a surface oxidation modulated trap distribution led to enhanced space charge injection and bulk electrical field distortion, which decreased DC breakdown strength as the oxidation duration went longer, in all film thicknesses. However, such decreases in breakdown strength occurred only in films below 55 μm under AC stresses, as the enhanced electrical field distortion at the electrode–dielectric interface was more obvious and dominating in thin films. These experimental results further confirm the proposed electrode–dielectric interface breakdown of dielectric films and provide new understandings of space charge modulated electrical breakdown, which fulfills dielectric breakdown theory and benefits the miniaturization of power equipment and electronic devices.

## 1. Introduction

Low-density polyethylene (LDPE) is a semi-crystalline polymer with crystallinity in the range of 50–60%, providing the LDPE with a good breakdown performance, excellent mechanical processing performance and resistance to harsh environments. Moreover, LDPE is easy to obtain with a relatively low cost, which promotes its wide applications as a typical insulating material in power equipment. However, insulating materials may be subject to ozone oxidation due to the ionization of oxygen in the air during fault conditions, such as corona discharge [1,2]. Therefore, it is necessary to study the effect of ozone oxidation on the structure and breakdown properties of LDPE.

Surface oxidation is currently widely applied in polymer industries to modulate polymer surfaces. Generally, an ozone generator applies corona discharge to produce ozone. In more detail, when electrons collide with oxygen molecules, the oxygen molecules are excited and eventually produce ozone after a series of reactions. At the same time, the heat generated by the electric current through the air causes the dissociation of oxygen molecules, producing free oxygen atoms, which combine with other oxygen molecules to form ozone. When polymers are exposed to ozone, hydroperoxides are formed in addition to the formation of carbonyl and carboxyl groups. These hydroperoxides have the ability to initiate the polymerization of vinyl monomers, resulting in grafting reactions on the surface of ozonated polymeric materials [3]. Compared to other surface treatments, surface ozone treatment exhibits the advantages of having a low cost and simple operation process and being easily adapted to complex surfaces, which indicate promising commercial use for surface ozone treatment in power equipment and electrical engineering.

Breakdown is an important phenomenon which occurs in electrical equipment and electronic devices. In power systems, electric energy is usually transmitted through AC and DC methods. However, the breakdown characteristics of typical insulating materials are extremely different under AC and DC voltages [4,5,6], and it is acknowledged that the DC breakdown strength is about twice or higher than the AC breakdown strength [7]. In the past decades, several breakdown theories have been proposed, such as thermal breakdown (proposed by Wager in 1922 [8]), electrical breakdown (proposed by Hipple in 1937 [9] and Frohlich in 1939 [10]), electrical-mechanical breakdown (proposed by Stark and Garton in 1955 [11]) and free volume breakdown (proposed by Artbauer in 1965 [12]). However, none of these theories can explain the typical breakdown phenomenon in different conditions. With the rapid development of electrical and electronic engineering, the incomplete mechanism of DC and AC breakdown has tremendously restricted the development of power equipment and electronic devices [4,13,14,15].

In recent years, space charges have been considered to play an important role in the breakdown and aging of polymeric insulating materials [16,17]. It has been observed that space charge accumulation and transportation can cause local electrical field distortion and trigger breakdown in DC conditions, proved with a pulsed electro-acoustic (PEA) method [18]. However, space charge profiles under AC electrical fields are difficult to experimentally observe [19], and thus, the phenomenon of AC breakdown related to the presence of space charges has not been clearly revealed until now. To further study the relationship between AC breakdown and space charges, Li et al. numerically simulated the space charge distribution and transportation characteristics in solid–liquid biphasic dielectrics under DC and AC (50–1000 Hz) conditions, showing that, under AC stresses, space charges are mainly accumulated in a very narrow depth within 0–2 μm near electrodes, and no charges are observed in the bulk of the material [20]. Such a narrow space charge distribution results in a very large electrical field distortion at the interfacial region, which initiates AC breakdown near the interface. The above investigation proves the role of space charges in triggering AC breakdown via a numerical simulation [20]. However, there is still a lack of further experimental evidence that shows the relationship between space charges and AC breakdown.

In this work, LDPE films with different thicknesses ranging from 30 μm to 230 μm were surface treated in an ozone atmosphere for 0 h, 1 h, 2 h, 4 h and 6 h to control the space charge profile, which is considered to be the dominating factor of DC and AC breakdown performances. The effects of surface oxidation on surface chemical structures, surface trap distributions and space charge accumulation of LDPE films before and after ozone treatment were measured in advance before analyzing the breakdown differences between AC and DC stresses.

## 2. Materials and Methods

### 2.1. Materials

LDPE (LE4147) in this article was purchased from Borealis, Vienna, Austria. This LDPE has a melt flow rate of 2 g/10 min at 190 °C/2.16 kg, a density of 0.918 g/cm^−3^ and a melting temperature of 109 °C.

### 2.2. Sample Preparation

In the experiments, circular LDPE films with a diameter of 6 cm and varied thicknesses (33 μm, 48 μm, 55 μm, 80 μm, 125 μm, 200 μm and 230 μm) were first formed via thermal pressing under 15 MPa for 15 min. Then, the LDPE films were evenly suspended in the reaction kettle, and then the reaction kettle was vacuumed until the vacuum in the reaction kettle dropped to −0.1 MPa. Subsequently, the outlet valve of the oxygen cylinder and the inlet valve of the reactor gas were opened. The ozone generated by the ozone machine entered the reaction kettle with the LDPE films suspended, and then the surface oxidation of the films was carried out. The concentration of ozone can be controlled by regulating the flow of oxygen into the ozone machine. In this experiment, films were placed in a reaction kettle filled with ozone under 0.04 MPa and 110 mg/L at room temperature for varied durations (0 h, 1 h, 2 h, 4 h and 6 h) to complete the surface modification procedure. Figure 1 is a schematic of the surface ozone treatment system.

### 2.3. Characterizations

#### 2.3.1. Fourier Transform Infrared Spectroscopy (FTIR)

For surface characterization, the chemical structure and composition were measured with the infrared spectrum (IR). The Thermo Fisher IN10 + IZ10 (Waltham, MA, USA) was used to measure the infrared spectra of each specimen at 500~3500 cm^−1^.

#### 2.3.2. Surface Potential Decay (SPD)

The surface trap distribution can be obtained with a surface potential decay (SPD) experiment. In the experiments, the distance between the surface potential probe (P0865, Trek, Lockport, NY, USA) and the film’s surface was maintained at 3 mm. During the tests, the charge voltage and gate voltage were –10 kV and –5 kV, respectively, and the charging time was 5 min. The temperature was kept at 30 °C, and the humidity was kept at about 30%RH.

#### 2.3.3. Volume Resistivity Tests

A three-electrode system was used to test the resistivity of LDPE films. The DC high-voltage power supply was provided by Spellman’s SL40PN300220 model (Hauppauge, NY, USA) with a maximum output voltage of 40 kV and a ripple factor of 0.1%. The ammeter that was used was Keithley’s 6517B electrometer (Beaverton, OR, USA), and the test range was 10 aA~21 mA.

#### 2.3.4. Pulsed Electro-Acoustic (PEA) Method

The space charge distribution in the films can be measured with the PEA method [21]. During the experiment, the pulse source voltage was 300 V, and the pulse source frequency was 150 Hz. The applied DC field strength was 50 kV/mm, and the applied voltage at each field strength lasted for 20 min.

#### 2.3.5. AC and DC Breakdown Tests

The AC and DC breakdown characteristics were tested with the HJC-100 kV breakdown tester (Huayang instrumentation Co., Ltd., Yangzhong, China). The electrode system was a standard sphere–sphere electrode made of brass with a diameter of 25 mm. During the tests, the film was sandwiched between two electrodes and then placed in transformer oil to prevent flashover. In this experiment, the breakdown voltage of the film was tested with the continuous boost method, and the voltage was boosted from 0 kV at a boost rate of 5 kV/s until the breakdown occurred. For each of the LDPE film thicknesses, 15 tests were conducted, and the Weibull distribution was used to analyze the breakdown characteristics.

## 3. Results

### 3.1. Chemical Structure

The IR results of ozone-treated LDPE films under varied durations are shown in Figure 2. The series of peaks at 2750–3000 cm^−1^ are due to the stretching vibration of alkyl groups -CH_3_, -CH_2_ and -CH, and the peaks at 1450 cm^−1^ are due to the bending vibration of these groups. The peaks at 720 cm^−1^ correspond to the vibration of back bone. It is worth noting that a new absorption peak emerged at 1710 cm^−1^ (C=O) after ozone oxidation, and its strength gradually increased with the extended oxidation time. In the mechanism, ozone is a strong oxidant that has high reactivity [22], and polymers can be easily oxidized when they are exposed to the ozone atmosphere [23,24]. In the process of oxidation, ozone molecules attack the polymer surface, which can be verified by FTIR results through the replacement of C-H by a C=O group and by the gradually increasing number of C=O groups with the extension of the surface treatment duration.

### 3.2. Trap Distributions

The surface chemical composition of LDPE films is modulated by oxidation, which could result in varied trap distributions, and it is first characterized via surface potential decay (SPD) experiments, as shown in Figure 3. As observed in Figure 3a, the SPD curves became faster with the increasing ozone treatment duration for electrons, which indicates that surface trap trends were shallower, and the trapped charges were easier to detrap. To further confirm this consideration, the surface trap distribution was calculated using a modified Simmon’s theory [25], and the results are shown in Figure 3b for electron traps. It indicates that two trap levels existed in the LDPE films. In our past research, it was investigated that the deep and shallow trap levels gradually decrease with increasing oxidation durations for holes. In addition, with the growth of the oxidation duration, the deep trap density decreases, whereas the shallow trap density increases [26]. The shallowed average trap depth and the increased trap density indicate that ozone treatment introduced more shallow traps in the surface layer of the LDPE film. Although the average trap energy level in Figure 3b does not change significantly, the shallow trap density increased during oxidation treatment, which indicates that oxidation treatment can also increase the shallow trap density for electrons in the surface layer of LDPE.

To clearly demonstrate the surface oxidation-dominated trap density, variations in shallow and deep traps were extracted from Figure 3b for electrons, as shown in Figure 4. The deep trap density decreased, and the shallow trap density increased along with the increase in oxidation duration. Generally, the total trap density increased due to the stronger increase in shallow trap density. This implies that surface oxidation exhibits effects on generating shallow traps for electrons.

Statistics on the functions of surface oxidation on trap information are shown in Table 1. With the increase in oxidation duration, both the depth of shallow and deep traps decreased. In addition, the shallow trap density increased, whereas the deep trap density decreased. Such decreases in average trap depth could lead to decreased electrical insulation characteristics in general, and this consideration was further investigated through a volume resistivity test, the results of which are shown in Figure 5.

In Figure 5, with the increase in oxidation duration, the volume resistivity gradually decreased. For pure LDPE, its volume resistivity was 2.85 × 10^17^ Ω·cm, and its volume resistivity decreased to 2.93 × 10^16^ Ω·cm after 6 h of surface oxidation. For the relationship between resistivity and carriers, a formula is generally acknowledged as follows: *ρ* = 1/(nq*μ*)(1)
where *ρ* is the resistivity of the materials, n is the number of carriers, q is carrier charge, and *μ* is the carrier mobility. The surface ozone treatment lowered the charge injection barrier, resulting in more free charge injection. With the increase in oxidation time, the number of surface carriers increased, which could affect charge injection and migration in the bulk of the film, triggering a decline in the volume resistivity. Moreover, the decline in volume resistivity was due to the decreased trap level, as confirmed by the SPD results. The decreased trap level caused a weakened trapping capture capability, which promoted carrier mobility, resulting in a decrease in the volume resistivity. This indicates that the surface oxidation treatment can reduce the bulk insulation properties of polymer materials. The decreased electrical insulating characteristics could result in a decreased electrical breakdown strength, which was investigated afterward. Space charges affected electrical breakdown by modulating electric field distributions. Here, the electrical field distributions of LDPE films after surface oxidation were investigated through the pulsed electro-acoustic (PEA) method, as shown in Figure 6.

Figure 6 shows the space charge distribution of LDPE oxidation films at 50 kV/mm, and huge differences in the space charge distribution between untreated and treated LDPE films can be observed. The space charge distribution of LDPE was dominated by the accumulation of positive heteropolar charges near the cathode, and the number of heteropolar charges increased gradually with increasing oxidation durations. Also, the positive homopolar charges at the anode gradually accumulated, indicating that there was a significant charge injection at the anode. For the oxidation-treated films (taking the OL-2 oxidation treatment for 2 h as an example), a large number of positive heteropolar charges were also accumulated at the cathode, but the space charge at the anode gradually changed from a negative heteropolar charge to a positive homopolar charge, which was significantly different from LDPE without the oxidation treatment.

Moreover, the injection intensity of the space charge gradually increased with the increase in oxidation time, but long-term oxidation had little effect on the electrode, which could introduce more heteropolar space charge, leading to a more complicated electric field distortion. For example, the space charge amplitude at the cathode of the LDPE and OL-2 films at 50 kV/mm was 13 cm^−3^ and 70 cm^−3^, respectively, but the space charge amplitude at the cathode of the OL-6 film was 57 cm^−3^. The above analysis demonstrates that surface oxidation can affect the space charge injection and accumulation characteristics of polymer dielectrics. After the surface oxidation treatment of polymers, the space charge injection (especially the hole at the anode) from the electrode/dielectric interface was significantly enhanced. The injection intensity gradually became stronger with the increase in oxidation duration, but the effect was not obvious when the oxidation time was too long.

Figure 6d shows that the total amount of space charge within the film gradually increased as the oxidation time increased. When the oxidation time was less than 2 h, Q_total_ increased rapidly. When the oxidation time was over 2 h, the trend of Q_total_’s increase in the sample became slow. Q_total_’s calculations also coincided with the space charge distributions in the PEA results, where the amplitude of the space charge near the cathode of OL-2, OL-4 and OL-6 films was close (between 56 and 70 cm^−3^).

### 3.3. Breakdown Behavior

Such an enhanced space charge injection, complicated space charge distribution and electrical field distortion by surface oxidation will inevitably modulate DC and AC breakdown behavior, as demonstrated in Figure 7. Figure 7a,c indicate that both the DC and AC breakdown strengths of all LDPE films under varied ozone treatment durations decreased with the increase in film thickness. Additionally, Figure 7a,c add surface oxidation into thickness-dependent breakdown, introducing differences in breakdown strength as the surface oxidation duration was prolonged. A comparison between Figure 7a,c clearly demonstrates that, with the increased film thickness, differences in *E*_DC_ under varied oxidation durations were enlarged, whereas differences in *E*_AC_ were narrowed. Figure 7b shows that the DC breakdown strength (*E*_DC_) decreased with the increasing surface treatment duration, despite the film thickness. Figure 7d shows the AC breakdown strength (*E*_AC_) as the surface treatment duration changed, showing that *E*_AC_ barely changed with the increasing oxidation time when the film thickness was larger than 80 μm, and under 55 μm, an observable decrease could be found with the extending surface treatment duration. From the results above, two preliminary conclusions can be drawn. First, both the DC and AC breakdown strengths of LDPE films decrease with film thicknesses. Second, surface treatment leads to a decreased DC breakdown strength for films in all thickness ranges (as shown in Figure 7b), indicating that the influence of surface treatment on DC breakdown performance is not thickness-dependent. However, surface treatment did not affect the AC breakdown performance of thick LDPE films above 80 μm, which implies that the effect of surface treatment on AC breakdown is closely related to the film thickness.

## 4. Discussion

### 4.1. Thickness-Dependent Electrical Breakdown

For the thickness-dependent electrical breakdown, an inverse law is generally acknowledged as follows [27]:*E*(d) = *k*d^−n^,(2)
where *E* is the breakdown field, and *k* and n are constants that are associated with the material.

The above inverse law is empirical and obtained from quantities of breakdown experiments. In the past years, various attempts have been made to better understand the thickness dependence of materials under DC, AC and impulse stresses [28]. For instance, it is believed in early work that defects in dielectrics dominate the breakdown, and the number of defects in the material increases with the volume. Thus, the thicker sample presents a lower breakdown strength [29]. However, such a theory has been proved to be invalid by experiments [29]. After that, space charge injection and accumulation have been considered to be dominating during the breakdown process [30,31,32,33,34,35]. Chen proposed a charge dynamic model to explain the thickness-dependent breakdown of polymeric materials [27]. In the model, space charge dynamics and electric field distributions across the bulk of the material were systematically investigated, indicating that electrons and holes are separately injected from the cathode and anode, and the accumulating area of negative charges has more spread than positive charges. Thus, the electric field distribution across the sample is seriously influenced by the injected charges, and the maximum electric field occurs in the bulk of the sample, proving that DC breakdown is initiated from the inner part of the film [36,37].

To further understand thickness-dependent DC breakdown, Li further simulated space charge dynamics during the voltage ramping procedures of electrical breakdown, showing that more homo space charges are injected with increased film thicknesses. In addition, as the voltage ramping duration is lengthened, the electrical field distribution becomes more distorted, eventually resulting in a decreased breakdown strength [20]. With the obtained mechanism of DC breakdown, Li further investigated space charge dynamics under AC stresses and found that both positive and negative charges are accumulated in a very narrow depth (within 0–2 μm) near electrodes, and no space charge is accumulated inside the middle of the film [20]. Although the absolute amount of the accumulated charge is comparatively smaller under AC stresses, a tremendously distorted field is built near the sample–electrode interface because of the narrowed accumulating area. Based on the above simulation, interface breakdown under AC stresses is proposed. From the above description, it is concluded that the breakdown of dielectrics is mainly influenced and modulated by space charge accumulation, and DC and AC breakdown are initiated in bulk and at the film–electrode interface, respectively.

### 4.2. Surface-Oxidation-Dependent Electrical Breakdown

The DC breakdown results in Figure 7b show a clear decrease in the prolonged surface oxidation duration. The above analyses indicate that the total amount of space charge in the bulk of the film could greatly affect its breakdown performance, but clearly, it is not the only dominating factor. In the mechanism, as has already been described in Section 3.2, the function of space-charge-modulated electrical breakdown is characterized by electrical field distortion. Both the trap depth and trap number modify such electrical field distortion. It is inevitable that the increased trap number leads to more space charge accumulation, resulting in a more serious filed distortion. The decreased average trap energy through the surface oxidation process indicates that charges are not firmly captured by trap centers as compared with LDPE films without surface modification. With the decreased activation energy to transfer trapped charges to free charges, more charges could be involved in electrical conduction under a high electrical field [17]. Moreover, these charges more easily gain energy and accelerate to trigger charge multiplication during electrical breakdown, which finally results in a lower breakdown strength.

The above description explains the decrease in the DC breakdown strength of LDPE after surface ozone treatment despite increasing film thicknesses (results shown in Figure 7a,c). However, such a decrease only occurs in AC breakdown in thick films above 80 μm, whereas under 55 μm, surface oxidation treatment barely changes AC breakdown behavior, as demonstrated in Figure 7b,d. From our previous knowledge, AC and DC breakdown strengths change synchronously with the treatment or modification method. For instance, both AC and DC breakdown strengths of nanofiller-enhanced dielectrics increase initially and then decrease with the content of nano fillers [38,39,40,41,42,43,44,45,46]. However, it seems that such a synchronous change does not exist when modifying polymeric films with a surface treatment. Zhao obtained similar results of decreased DC breakdown strengths and unchanged AC breakdown strengths in LDPE after surface fluorination [47].

In Li’s simulation, it was already proposed that the DC and AC breakdown mechanisms are bulk and interface breakdown, respectively [20]. However, this lacks further experimental support. Our experimental results in this work provide strong evidence for AC breakdown originating from the electrode–material interface. In general, surface treatment only changes surface chemical structures within a thickness of several micrometers. However, such a thick surface treatment layer would have different influences on DC and AC breakdown strengths based on bulk breakdown and interface breakdown.

For the DC condition, the changes in charge traps with distorted electrical fields are verified by the PEA measurements in Figure 6. Surface oxidation results in more positive and negative charges injected into films and trapped in trap centers. The space charge profiles from both SPD and PEA show that surface oxidation results in a decreased trap depth and increased trap density. When charges are trapped in shallow traps, such trapped charges are not firmly immobilized and could transport to the counter electrode by detrapping and re-trapping processes under high electrical strength [16]. Thus, these injected charges can easily transport to the bulk of the film, and continuous charge injection leads to strong charge accumulation in the film bulk, which distorts the electrical field and results in a lower breakdown strength with all thicknesses.

For the AC condition (interface breakdown), the charges are accumulated in the very narrow thickness of 0~2 μm in the vicinity of electrodes, and the maximum electric field is concentrated in this interface area. This means that this interface region undergoes breakdown first, and then the next layer undergoes breakdown until the whole breakdown occurs. When the film is relatively thick, the thickness ratio of the surface modification layer is very small, and the breakdown of the surface modification layer cannot have an obvious influence on the rest of the untreated film. Thus, the breakdown strength is not changed for surface-treated thick films. But when the film is relatively thin, the breakdown of surface layer modification has an obvious effect on the rest of the untreated sample. When the surface layer breaks down, the whole voltage is applied to the untreated parts, which results in the electric field being distorted seriously in this area, and breakdown occurs subsequently.

## 5. Conclusions

The space charge behaviors and AC and DC breakdown mechanisms of surface-ozone-treated LDPE films were investigated via experiments.

With the extension of the oxidation duration, the shallow trap density for electrons and holes increased, but the average trap level reduced.Space charge injection from the electrode–dielectric interface was significantly enhanced after surface oxidation, and the injection strength gradually became stronger with the increase in oxidation treatment duration.The DC and AC breakdown strengths of LDPE films decreased with the increase in film thickness during the oxidation treatment due to the stronger homo charge injection.For DC breakdown, surface oxidation introduced shallow traps, resulting in more involved charges in electrical conduction, which more easily gained energy and accelerated to trigger charge multiplication, which finally resulted in a lower breakdown strength.The effect of oxidation on AC breakdown is closely related to film thickness. For thick films, the breakdown of the surface-modified layer cannot have a significant effect on the rest of the untreated film, leading to a barely changed breakdown strength with changes in film thickness.For surface oxidation, an ozone concentration up to 110 mg/L and an oxidation time up to 6 h are recommended, and it is optimal to stabilize the pressure in a reaction kettle at 0.04 MPa. A minimum thickness of 10 μm is recommended for polymeric insulating materials, since thinner films might be vertically penetrated during surface oxidation.

## Figures and Tables

**Figure 1 polymers-15-04490-f001:**
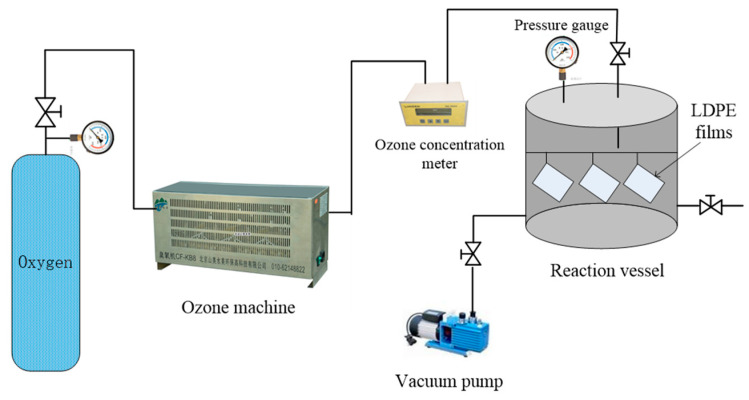
Schematic of surface ozone treatment system.

**Figure 2 polymers-15-04490-f002:**
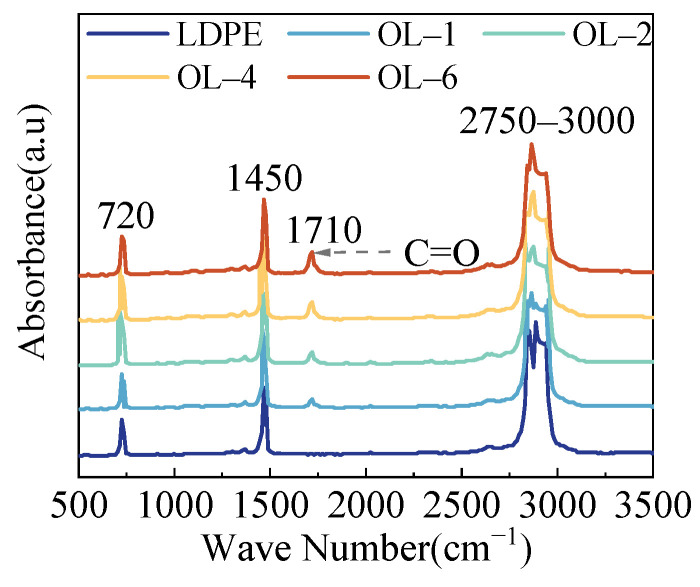
IR spectrum of LDPE films under ozone treatment.

**Figure 3 polymers-15-04490-f003:**
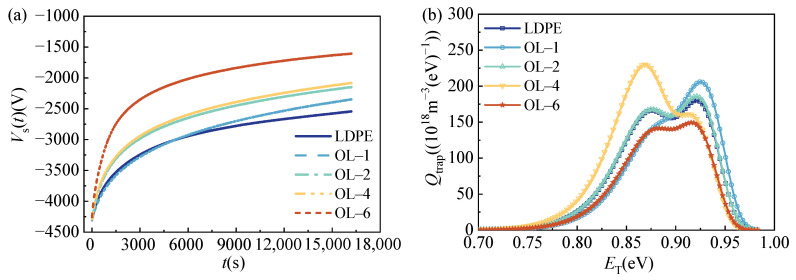
SPD characteristics and surface trap distributions of LDPE films for electrons. (**a**) SPD curves of LDPE films for electrons. (**b**) Surface trap distributions of LDPE films for electrons.

**Figure 4 polymers-15-04490-f004:**
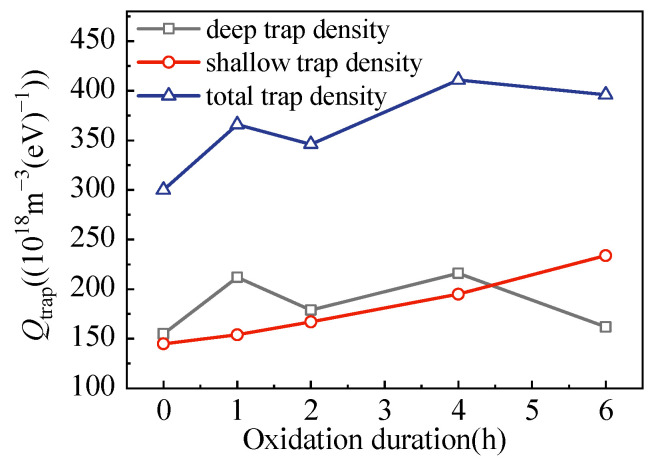
Trap density of LDPE films for electrons.

**Figure 5 polymers-15-04490-f005:**
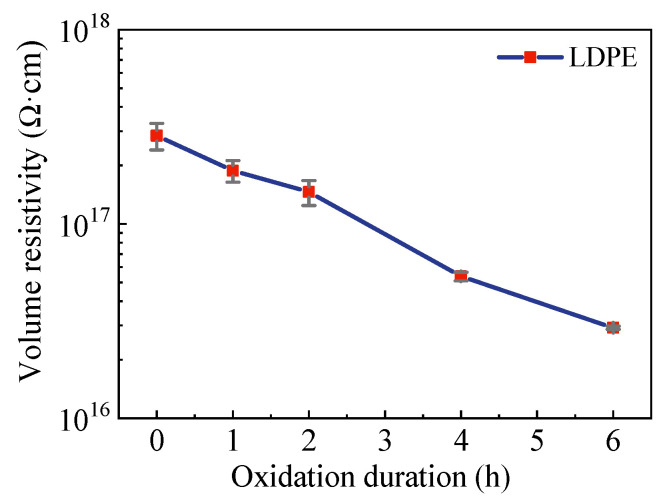
Effects of surface oxidation duration on LDPE films’ volume resistivity.

**Figure 6 polymers-15-04490-f006:**
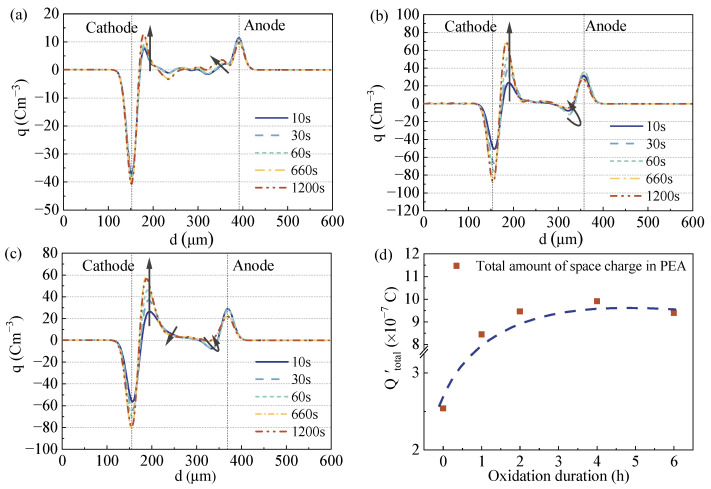
Space charge distribution of LDPE films in PEA at 50 kV/mm. (**a**) Space charge distribution of LDPE in PEA at 50 kV/mm. (**b**) Space charge distribution of OL-2 in PEA at 50 kV/mm. (**c**) Space charge distribution of OL-6 in PEA at 50 kV/mm. (**d**) Effects of oxidation duration on LDPE films’ total amount of space charge (Q_total_) (The blue dotted line represents the calculations of Q_total_).

**Figure 7 polymers-15-04490-f007:**
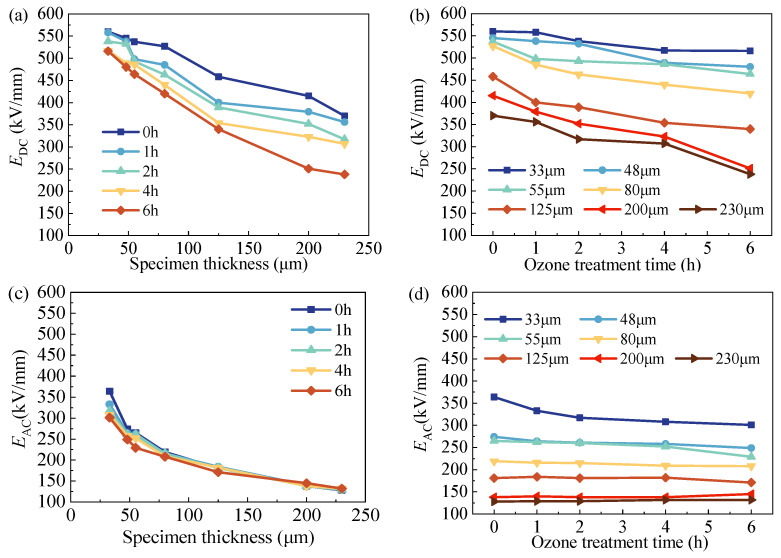
*E*_DC_ and *E*_AC_ of all LDPE films under varied ozone treatment durations. (**a**) *E*_DC_ of surface-treated LDPE films as a function of film thickness. (**b**) *E*_DC_ of LDPE film thickness as a function of surface treatment duration. (**c**) *E*_AC_ of surface-treated LDPE films as a function of film thickness. (**d**) *E*_AC_ of LDPE film thickness as a function of surface treatment duration.

**Table 1 polymers-15-04490-t001:** Surface trap depth and trap density of LDPE films.

Films	Trap Depth (eV)		Trap Density (×10^18^ m^−3^(eV)^−1^)	
	Shallow	Deep	Shallow	Deep
LDPE	0.88	0.93	142	280
OL-1	0.87	0.92	168	206
OL-2	0.86	0.91	216	224
OL-4	0.86	0.91	289	193
OL-6	0.86	0.90	342	141

## Data Availability

The data presented in this study can be obtained upon request from the corresponding authors.

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
