# Peer review of "Investigation on AC and DC Breakdown Mechanism of Surface-Ozone-Treated LDPE Films under Varied Thicknesses"

_polymers, 2023, doi:10.3390/polym15234490_

Round 1
Reviewer 1 Report
Comments and Suggestions for Authors
The manuscript shows interesting study of breakdown mechanism of LDPE films. Large experimental data was obtained, however there is some lack of justification in the article which makes it not clear enough for readers and decreases significance of the results.
Please, revise the manuscript taking into account the following comments:
1. The introduction section should be significantly enlarged to show the context clearer.
2. At least brief description of LDPE material is needed along with explanation why it was chosen for the experiments.
3. What was the size of films? Does it matter and affect on results?
4. Based on information given in the introduction, it’s not clear what for ozone treatment was performed. What is the aim of the treatment in context “experimental evidence that shows the relationship between space charge and AC breakdown”?
5. How can experimental data of the effect of surface oxidation help in practice? Examples of possible practical application should be provided. Now it is not clear enough, it is difficult to assess the significance of the results obtained.
6. It seems that recommendations about optimal film thickness and/or ozone treatment parameters for various working conditions should be given.
Reviewer 2 Report
Comments and Suggestions for Authors
The paper presents the study about investigation on AC and DC breakdown mechanism of surface ozone treated LDPE films under varied thicknesses. Authors investigated LDPE films with various thicknesses, were surface was modified by ozone for different durations, for experimentally investigating DC and AC breakdown mechanisms. The obtained results prove that carbonyl groups were introduced onto the film surface, forming shallow surface traps, leading to decreased average trap depth and increased trap density. These experimental results confirm the proposed electrode-dielectric interface breakdown of dielectric films and provide new understandings of space charge modulated electrical breakdown, which fulfills dielectric breakdown theory.
Dear author, thank you very much for interesting paper about the increase of the theory of AC and DC breakdown mechanism. I put some comments and question.
Comments:
1. The introduction is well written. Authors remembered the theory about the breakdown, such as electrical breakdown, thermal, mechanical, etc.
2. Authors did not explain enough the influence of time on volume resistivity, on Fig. 5. Please explain physical phenomenon what happen in investigated material using time, if possible.
3. Fig.7. – what international procedure was used to measure DC and AC breakdown voltage?
4. Fig.7. – I suggest to use the same range of the scale for all four subpictures a-d, if possible.
5. Fig.7. – a, c – it is typically that with the increase of thickness, breakdown field decreases. What is new? Please explain if possible.
6. Summarizing, the paper results are interesting and value. They add new knowledge to explain the breakdown phenomenon in case of DC and AC voltage. Please complete some information I asked.
Round 2
Reviewer 1 Report
Comments and Suggestions for Authors
I would like to thank authors for thorough revision and detailed answers for given comments.
I am satisfied with them, however, there is no corrections in the Conclusions, although large description about possible recommendations was provided in cover letter. I do recommend to add main points and suggestions from cover letter to the Conclusions section.
